# Physician consultation in young children with recurrent pain—a population-based study

G Hirschfeld[1,2], J Wager[2,3] and B Zernikow[2,3]

[1] Faculty of Business Management and Social Sciences, University of Applied Sciences Osnabrück, Osnabrück, Germany
[2] German Paediatric Pain Centre, Children's Hospital Datteln, Germany
[3] Department of Children's Pain Therapy and Paediatric Palliative Care, Faculty of Health, School of Medicine, Witten/Herdecke University, Germany

Corresponding author
G Hirschfeld,
hirschfeld@wi.hs-osnabrueck.de

## ABSTRACT

**Background.** Recurrent pain is a common experience in childhood, but only few children with recurrent pain attend a physician. Previous studies yielded conflicting findings with regard to predictors of health care utilization in children with recurrent pain.

**Methods.** The present study analyzes data from the German Health Interview and Examination Survey for Children and Adolescents (KiGGS) study comprising $n = 2{,}149$ children (3–10 years old) with recurrent pain to find robust predictors. We used multiple logistic regressions to investigate age, gender, socio-economic status (SES), migration background, pain intensity, pain frequency, pain-related disability, mental health problems, and health-related quality of life (HRQL) as predictors for visiting a doctor due to pain.

**Results.** Overall, young girls with high pain-related disability, intensity, frequency, and migration background were more likely to attend a physician. Pain-related disability had the largest impact. Socioeconomic status, health-related quality of life and mental health problems were not systematically related to health care utilization. An analysis of the variability of these results indicated that several hundred participants are needed until the results stabilize.

**Conclusions.** Our findings highlight the importance of pain-related disability and frequency in assessing the severity of recurrent pain. Generic predictors and demographic variables are of lesser relevance to children with recurrent pain. On a methodological level, our results show that large-scale studies are need to reliably identify predictors of health care utilization.

## INTRODUCTION

Recurrent pain in children and adolescents is a pervasive health care issue but only between 36% and 54% of the children with chronic pain visit a physician (*Perquin et al., 2001*; *Ellert, Neuhauser & Roth-Isigkeit, 2007*; *Huguet & Miró, 2008*). Over the last ten years a number of studies aimed to describe those children who not only have recurrent pain but also visit

physicians or specialized treatment centers (*Peng et al., 2007*; *Wager et al., 2013a*). Factors associated with physician visits due to recurrent pain may be relevant to better understand children seeking professional help, and shape the health care system.

So far, only three studies have systematically investigated pain- and demographic characteristics as predictors of health care utilization in pediatric pain patients (*Perquin et al., 2001*; *Huguet & Miró, 2008*; *Toliver-Sokol et al., 2011*). These studies agree that pain characteristics are associated with pain-related health care utilization. Children with continuous pain, high pain frequency and high pain intensity are more likely to visit a physician (*Perquin et al., 2001*; *Huguet & Miró, 2008*). Furthermore, interference with daily activities and school absence due to pain are factors associated with health care utilization (*Perquin et al., 2001*; *Huguet & Miró, 2008*; *Toliver-Sokol et al., 2011*). These studies found conflicting findings concerning demographic characteristics. While one study indicates that girls with chronic pain seek health care more often than boys (*Perquin et al., 2001*), other studies did not replicate this gender difference (*Huguet & Miró, 2008*; *Toliver-Sokol et al., 2011*). Similarly, one population-based study found that younger children seek a physician more often (*Ellert, Neuhauser & Roth-Isigkeit, 2007*), the other studies did not replicate these findings (*Perquin et al., 2001*; *Huguet & Miró, 2008*). Some generic variables, e.g., comorbid somatic diseases (*Perquin et al., 2001*) or health-related quality of life (HRQL: *Huguet & Miró, 2008*) have only been studied once. Even though emotional distress is associated with recurrent pain (*Skrove, Romundstad & Indredavik, 2014*), so far no study investigated the role of emotional distress for health care utilization in children with recurrent pain. The adult literature suggests that emotional distress accounts for more health care utilization in chronic pain (*Hirsch et al., 2014*). Furthermore, variables that are highly relevant predictors of health care utilization in other contexts such as socioeconomic status (SES) or migration background, have rarely been studied, and if have been found to not be related to health care utilization (*Perquin et al., 2001*; *Toliver-Sokol et al., 2011*). Reasons for these inconsistencies may be small sample sizes ($N$) relative to the number of variables studied ($K$), and use of univariate analysis (*Huguet & Miró, 2008*). One study (*Perquin et al., 2001*) tested $K = 28$ variables in a sample comprising $N = 254$ children with chronic pain from a larger epidemiologic study, the other study used multiple linear regression analysis to test $K = 6$ variables in $N = 59$ patients (*Toliver-Sokol et al., 2011*). Such a small number of cases per variable may result in spurious findings and overall highly variable effect-estimates.

The aim of the present research is to identify robust predictors of health care utilization in children with recurrent pain in a large population-based study. Specifically, we used a multiple logistic regressions approach to investigate age, gender, SES, migration background, pain intensity, pain frequency, pain-related disability, mental health problems and HRQL, as predictors for visiting a doctor due to pain. The secondary aim is to describe the variability of such estimates and determine the sample size that is needed to yield stable estimates for the predictors.

## MATERIALS AND METHODS

### Subjects

We analyzed a subset of the participants of the German Health Interview and Examination Survey for Children and Adolescents (KiGGS). The KiGGS was approved by the Charité/Universitätsmedizin Berlin ethics committee and the Federal Office for the Protection of Data. The KiGGS study gathered nation-wide data on health in children and adolescents in Germany. Between May 2003 and May 2006, data from 10,333 children between 0 and 10 years and 7,308 adolescents between 11 and 17 years were gathered in 167 cities and towns throughout Germany. KiGGS consists of different modules for various domains of well-being including health-related quality of life and pain (*Kurth et al., 2008*). We analyzed only a subset of the children whose parents reported (1) the child had pain at least once per month in the past three months and (2) data was available on all predictor variables (see below). Because the prevalence for pain varies across gender and age, this study sample is systematically different to the full sample (*Ellert, Neuhauser & Roth-Isigkeit, 2007*). Since there are no systematic biases that prevent children in pain from participating in KiGGS, we believe that the sample analyzed in the present manuscript is representative for children with recurrent pain.

### Measures

As part of the KiGGS study parents answered questions regarding demographic information and different aspects of pain in the preceding three months. Data were based on parental report. Health care utilization was assessed with one item asking whether the child visited a doctor due to his/her pain problem. The five response categories ranged from "never" to "always". For the present analysis these were dichotomized by collapsing all responses from "once" to "always" so that the analysis compared those who attended a physician at least once to those who never attended a physician. While this reduces the amount of variance this is in line with previous studies that also used this contrast to define predictors (*Perquin et al., 2001*; *Huguet & Miró, 2008*).

We used demographic, pain-related and generic variables as predictors of health care utilization in children. Demographic variables were age, gender, SES, and migration background. The SES includes information on the parental education, their occupation as well as the family income (*Lange et al., 2007*). It was divided into "high", "middle" and "low". Migration background was defined as either (1) the child being an immigrant and at least one parent not being born in Germany or (2) both parents being immigrants and being no German citizen (*Lange et al., 2007*).

We assessed several pain-related predictors for health care utilization; pain-related disability, pain-frequency, and pain-intensity. We calculated a comprehensive pain disability measure by adding up the impairment in different areas of everyday life including school, meeting friends, appetite, sleep, hobbies (*Hirschfeld & Zernikow, 2013*). Parents were asked to judge their children's pain-related impairment by indicating how often specific scenarios were met e.g., "child was not able to attend daycare because of their children's pain" and responded on a scale ranging from "never" (=1) impaired due to

pain to "always" (=5). Higher scores are indicative of higher impairment. Average pain intensity was measured with a 100 mm visual analogue scale. Pain frequency in the past three months was described on a scale ranging from "once a month" (=2) to "daily" (=6).

Furthermore, we assessed general predictors for health care utilization: HRQL, and mental health. HRQL was assessed with the KINDL-R (*Bullinger et al., 2008*). This 24-item questionnaire covers six dimensions of HRQL. For this study we used the scales "psychological well-being" and "physical well-being." Aspects of well-being were rated on a 5-point Likert-scale. The strengths and difficulties questionnaire (SDQ) (*Goodman, 1997*) is a screening instrument covering mental health problems. In KiGGs study the parent version of the SDQ was used for all participants. In the present analysis the total difficulties score was used, that was computed as the sum of the four difficulty scales; "Emotional symptoms," "Conduct problems," "Hyperactivity/inattention" and "Peer relationship problems."

## Data analysis

Data were analyzed in two levels: the full sample and the pseudosamples that were generated as part of the sequential sampling approach. In the full sample, predictors of health care utilization were identified using multiple logistic regression analysis. In order to make the odds ratios (ORs) comparable across predictors measured on different scales, predictors were scaled by dividing by two standard deviations before being entered into the logistic regression (*Gelman, 2008*). This allows interpreting ORs for continuous predictors similar to ORs from binary predictors as the added—or reduced—risk of participants with higher scores compared to lower scores.

A sequential sampling approach was used to assess the variability of the parameter estimates. The sequential sampling approach tries to illustrate how the ORs of the predictors change when the sample size is gradually increased and identifies a sample size from which on these are stable. Specifically, this entails adding the participants one by one to the dataset and computing the logistic regression with each addition. The resulting sequence of ORs can be plotted against the sample-size showing the *trajectory* of the ORs. Some ORs for the individual predictors may either be relatively stable across the number of participants—e.g., the OR for pain-intensity in Fig. 2—while others may show some changes depending on the number of participants that were included in the analysis—e.g., the OR for migration in Fig. 2. Of special importance was the point of stability (POS), i.e., the number of participants that had to be included until the significance of this specific effect did not change any more or stabilized (*Schönbrodt & Perugini, 2013*). Based on the trajectory one can see that at low sample sizes adding participants affects the estimated OR much more than at larger sample sizes. Hence, adding more participants to the analysis before the POS may change not only the magnitude but also whether an effect is significant or not. After the POS, the effect remains stable, i.e., it remains either insignificant or significant. In the example below, the OR for migration was significantly smaller than one when only the first 100 participants were included and significantly larger than one when more than 500 participants were

| Table 1 Demographic and pain-related characteristics. | |
|---|---|
| **Domain** | **Analysis-sample Median ± iqr/N (%)** |
| Age (years: 3–10) | 7.41 ± 3.81 |
| Gender (female) | 1,156 (54%) |
| Migration (migrant) | 156 (7%) |
| Frequency | |
| • Once per month | • 563 (26%) |
| • 2–3 per month | • 894 (42%) |
| • Once per week | • 254 (12%) |
| • Several times per week | • 383 (18%) |
| • Daily | • 55 (3%) |
| Main-pain location | |
| • Abdomen | • 814 (37%) |
| • Head (incl. ears) | • 610 (29%) |
| • Back/Extremeties | • 463 (21%) |
| • Other | • 262 (12%) |
| Pain-Intensity (0–100 VAS) | 42.00 ± 31.00 |
| Pain-Disability [1–5 composite score] | 1.57 ± .83 |
| HRQL-Physical [0–100 KINDL] | 75.00 ± 25.00 |
| HRQL-Psychological (0–100 KINDL) | 81.25 ±12.5 |
| SDQ-Total (0–33) | 9.00 ± 7.00 |

**Notes.**

HRQL, Health related quality of life; SES, Socioeconomic status; SDQ, Strengths and difficulties questionnaire.

included. The POS for this effect in this specific sampling order was around 500 as from this sample size on adding participants did not change the sign and significance of the effect. Because trajectories and its' corresponding POS are specific to the particular order in which participants were added to the analysis, we replicated this analysis for 1.000 random orders of participants. For each of the 1.000 random orders, the trajectory and POS were calculated. This resulted in 1,000 slightly different POS for each OR. From the distribution of these POS we calculated the $POS_{crit}$ as the 80th percentile of the POS. This indicates the sample size from which on 80% of the ORs stabilized. Inspection of the POS for 1.000 sequences and $POS_{crit}$ provides an index for how many participants would have to be sampled before the solution stabilizes irrespective of the specific order in which the participants were sampled (*Schönbrodt & Perugini, 2013*). Data analysis was performed in R. Analysis scripts; the file "Public Use File KiGGS 2003-2006" may be requested at: http://www.rki.de/EN/Content/Health_Monitoring/Public_Use_Files/application/application_node.html will be made available upon request.

## RESULTS

### Patients

Data from 2,149 children (3–10 years; 7.41 years ± 2.25; 54% female) were analyzed. Table 1 gives an overview of the demographic and pain-related characteristics of the analyzed sample.

**Table 2 Results of the sequential sampling analysis.**

|  | POS | POS$_{crit}$ |
|---|---|---|
| Pain-related disability | 117 | 109 |
| Pain intensity | 224 | 414 |
| Pain frequency | 531 | 666 |
| Migration | 1,697 | 1,844 |
| Sex | 2,144 | 2,143 |
| HRQL-Psy. | 50 | 332 |
| SES | 50 | 301 |
| SDQ | 1,196 | 761 |
| HRQL-Phy. | 79 | 1,571 |
| Age | 701 | 902 |

**Notes.**

POS, Point of stability for the order presented in Fig. 2; POS$_{crit}$, Critical Point of stability across 1,000 random orders; HRQL, Health related quality of life; SES, socioeconomic status; SDQ, Strengths and difficulties questionnaire.

## Predictors of health care utilization in the full sample

Overall, we found that 1,144 (53%) of the children with recurrent pain consulted a physician due to their recurrent pain. Pain-related disability was the strongest predictor of whether or not a child would visit a physician due to recurrent pain (OR = 5.37 ; 95% CI [4.15–7.02]), children with high disability, due to pain were five times more likely to visit a physician (Table 2). As can be seen in Fig. 1, five other variables were also significant predictors but had much smaller ORs; pain-intensity (OR = 2.05; 95% CI [1.66–2.54]), pain-frequency (OR = 1.67; 95% CI [1.37–2.03]), migrant (OR = 1.64; 95% CI [1.12–2.41]), sex (OR = 1.22 ; 95% CI [1.01–1.47]), age (OR = .66; 95% CI [0.54–0.79]). Overall, young girls with high pain-related disability, -intensity, –frequency, and migration background had higher probabilities to visit a physician. Socioeconomic status, health-related quality of life and mental health problems were not systematically related to physician visits.

## Variability of predictor estimates

Figure 3 shows a trajectory of the estimates for one specific sampling order, i.e., the ORs from the logistic regression based on different subsamples of patients. In this sampling order the estimates of the effects change depending on how many participants are added to the analysis. The ORs for most effects converge to 1, i.e., a null-effect, in the first 250 participants. Critically the ORs may even reverse, e.g., the effect of migration background is smaller than 1 up until about 450 included participants, from which point on the effect becomes larger than 1.

Inspecting the POS for this specific order (Table 2) revealed that the effects of HRQL and SES did not change from the smallest sample size studied, i.e., the POS was 50. However, even the huge effect of pain-related disability was only stable from 117 participants onwards, and the effect of pain frequency needed 531 participants to stabilize. Of note three variables (Migration, Sex, and SDQ) needed over 1,000 participants until their effect could be reliably detected in this specific sampling order.

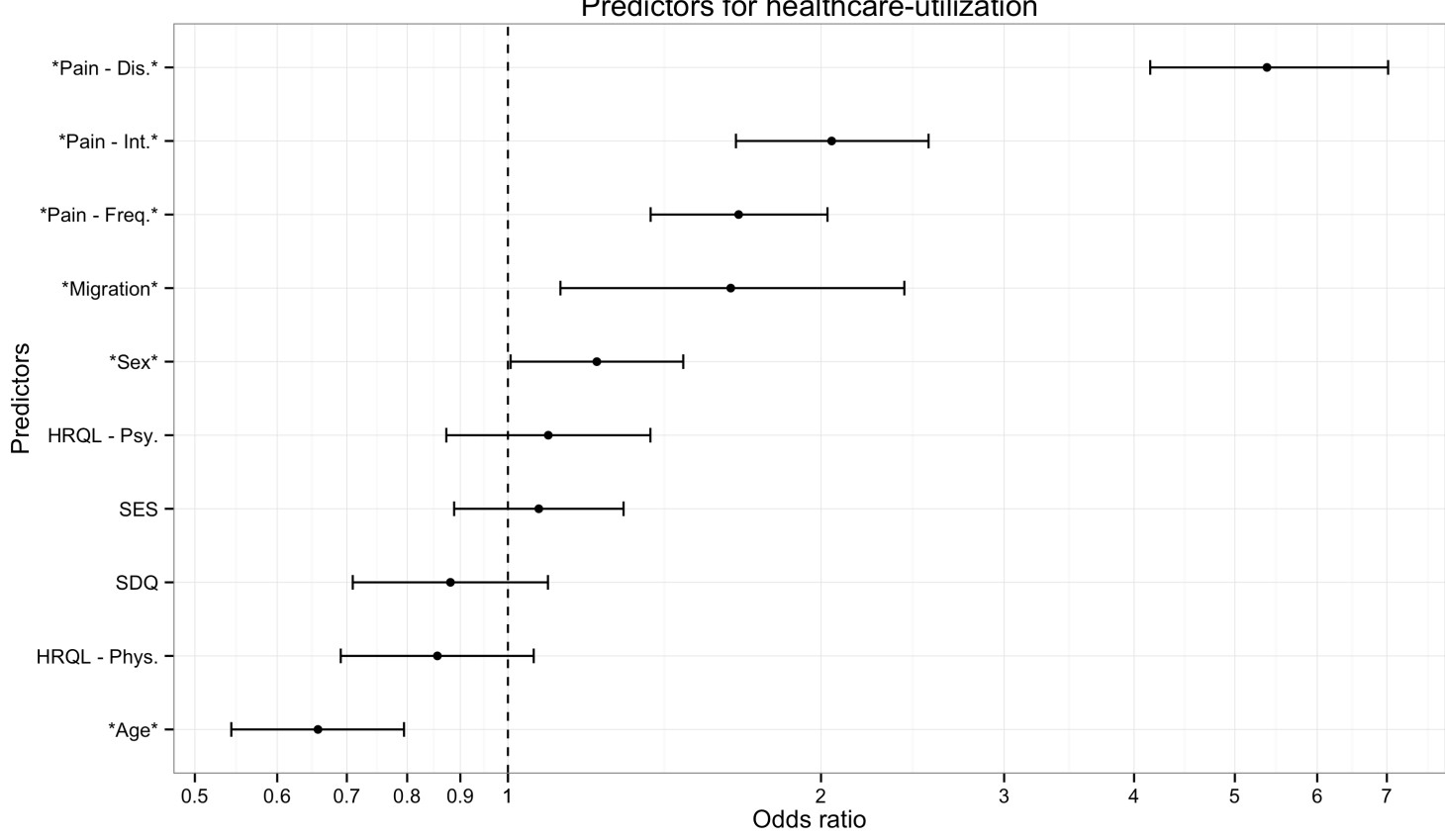

Predictors for healthcare-utilization

**Figure 1 Predictors of health care utilization.** The figure presents results of a logistic regression analysis predicting which children with recurrent pain will visit a doctor due to their pain. Odds-ratios indicate that young girls with high pain-related disability, intensity, frequency, and migration background were more likely to visit a physician. Socioeconomic status, health-related quality of life and mental health problems were not systematically related to physician visits. Note: Error bars indicate 95% Confidence intervals. Stars denote predictors that were significant in the full sample. Pain-Dis., Pain-related disability; Pain-Int., Pain intensity; Pain-Freq., Pain frequency; HRQL-Psy., Health related quality of life psychological; HRQL-Phys., Health related quality of life physiological; SES, Socioeconomic status; SDQ, Strengths and difficulties questionnaire.

Inspection of the $POS_{crit}$ from the 1,000 different random sequences (Fig. 3) demonstrates that the findings concerning the POS described above are not specific for the order in which the participants were sampled. The $POS_{crit}$ indicate that most effects stabilized only after several hundred participants were included in the analysis (Table 2). Of note, while the effect of HRQL-Phy. seemed to stabilize with few participants in the specific order studied above, this effect stabilizes relatively late when alternative orders are considered. Across the thousand different orders, the effect of pain-related disability stabilized with the fewest number of participants while the effect of sex, which was borderline significant in the full sample, stabilized extremely late.

## DISCUSSION

The aim of the present study was to identify predictors of health care utilization in children with recurrent pain. We found that pain-related disability was the largest and most robust predictor of health care utilization. Several other variables that predicted health

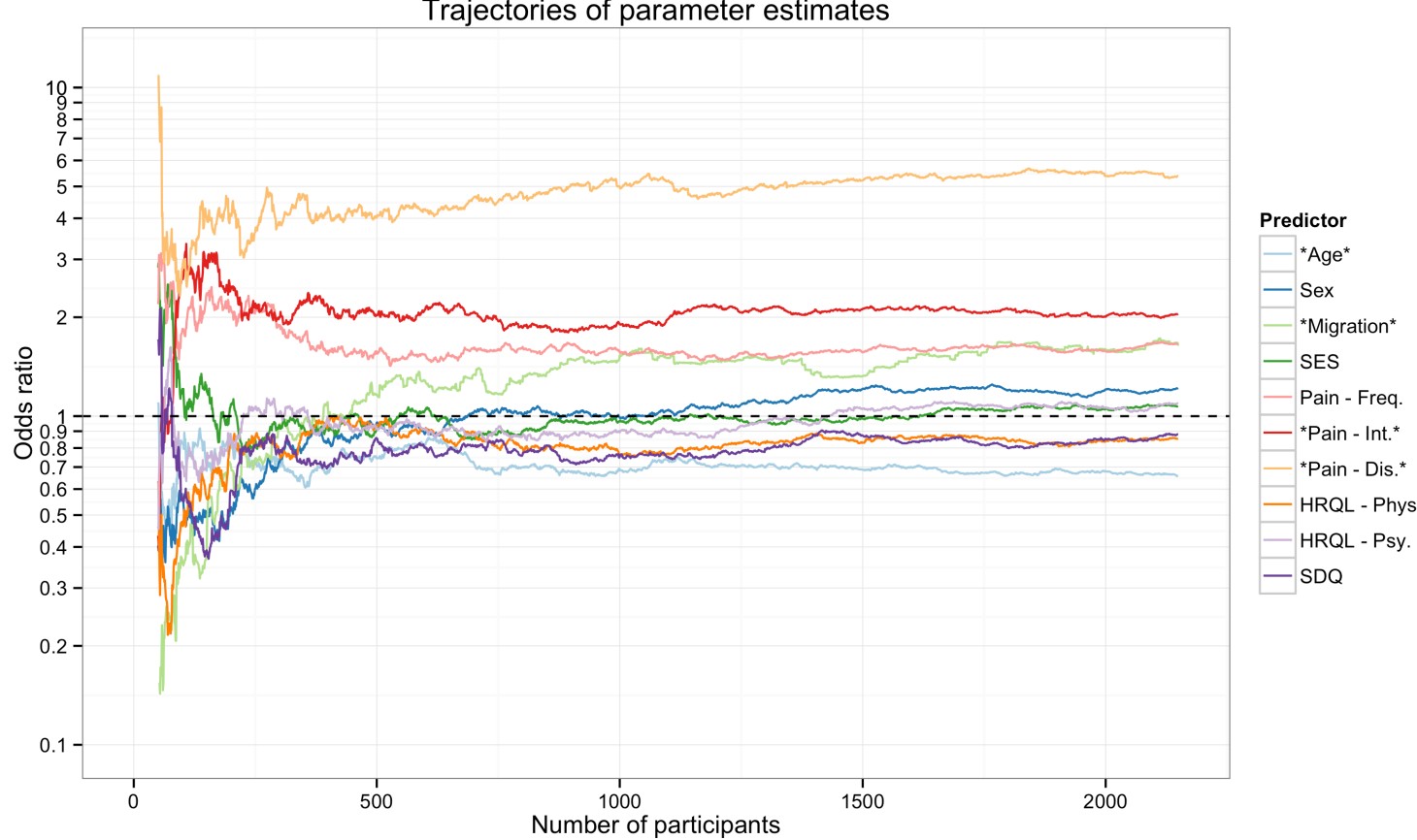

**Figure 2 Trajectory of the odds ratios for one sampling order.** The trajectory shown summarizes the odds ratios for the different predictors when participants were added one by one to the analysis. The points left points show the ORs for the first 50 participants and the points on the right show the ORs for 2,149 participants. Of note some effects change the direction, e.g., the effect of migration background is smaller than 1 up until about 450 included participants, from which point on the effect becomes larger than 1. Note: Stars denote predictors that were significant in the full sample. Pain-Dis., Pain-related disability; Pain-Int., Pain intensity; Pain-Freq., Pain frequency; HRQL-Psy., Health related quality of life psychological; HRQL-Phys., Health related quality of life physiological; SES, Socioeconomic status; SDQ, Strength and difficulties questionnaire.

care utilization in other contexts (SES, HRQL, SDQ) were not systematically related. An analysis of the variability of these results showed that half of the effects needed sample sizes of at least 666 participants, with larger effects requiring fewer and smaller effects requiring more participants. In what follows we will discuss the findings with regard to health care utilization before describing the general strengths and limitations of the study.

We found that pain-related disability and pain intensity are significant predictors of health care utilization. These findings support the assumption that not ongoing pain itself, but the suffering, e.g., in terms of pain-related disability, makes the difference between recurrent pain as a clinically relevant condition and as a private phenomenon. In line with our results, several previous studies have suggested these two factors as significant predictors for health care utilization (*Perquin et al., 2001*; *Huguet & Miró, 2008*; *Toliver-Sokol et al., 2011*). This finding underlines the need for a more holistic definition of clinically relevant pain. Most studies have tried to define meaningful levels of pain-intensity (*Hirschfeld & Zernikow, 2013*), less attempts have been made to use

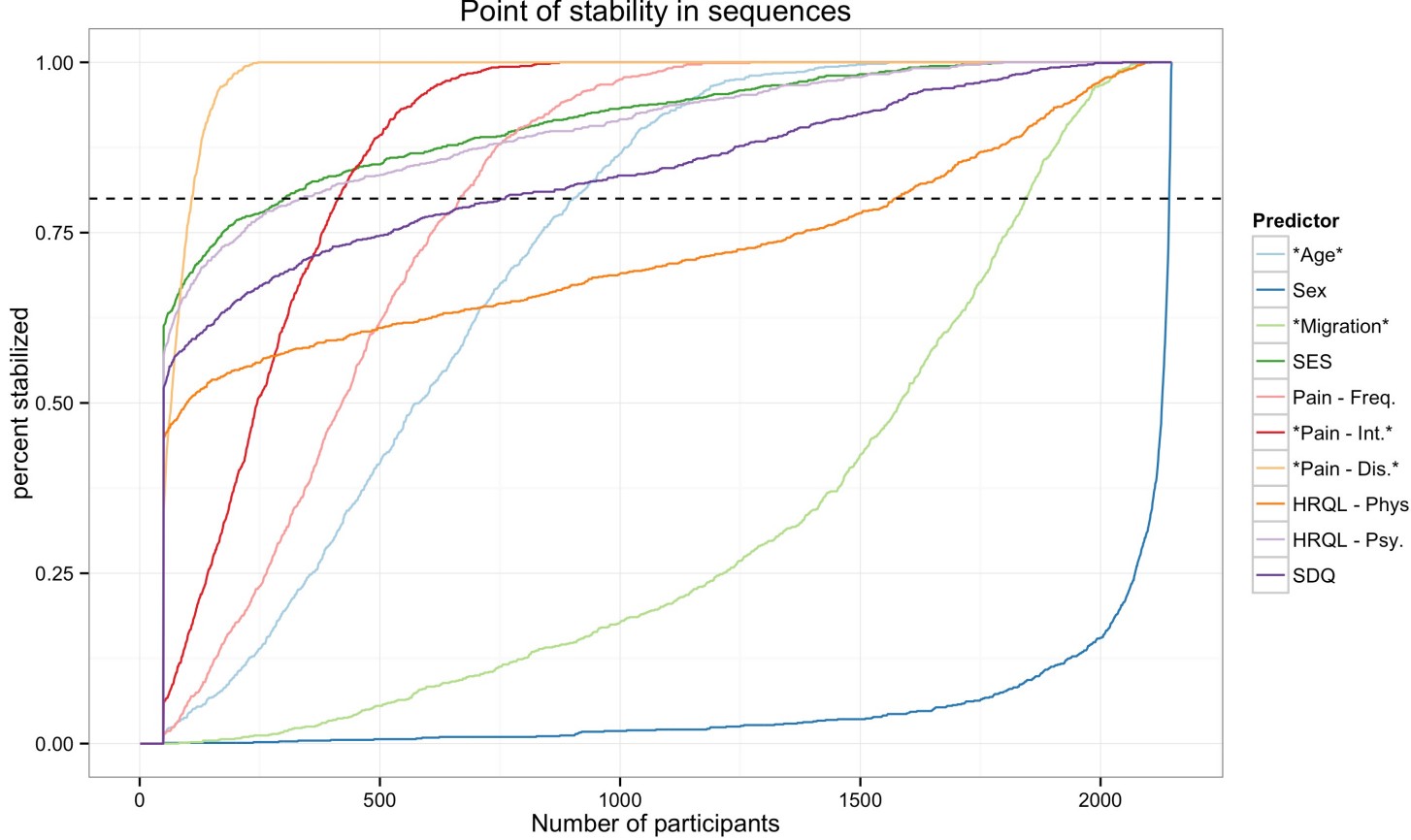

**Figure 3** **Point of stability for the parameter estimates.** The figure shows the cumulative frequency distribution of the POS for effect across 1,000 different random sequences. That is it shows how many of the 1,000 different orders have a POS that is smaller than the specific sample size. The higher the percentage of orders that have already stabilised the more confident can a researcher be that her results at this sample size are accurate and will not change with more participants. Note: Stars denote predictors that were significant in the full sample. Pain-Dis., Pain-related disability; Pain-Int., Pain intensity; Pain-Freq., Pain frequency; HRQL-Psy., Health related quality of life psychological; HRQL-Phys., Health related quality of life physiological; SES, Socioeconomic status; SDQ, Strengths and difficulties questionnaire.

criteria other than pain-intensity to describe the severity. Purves and colleagues (*1998*) have suggested to include a "recent or frequent seeking of treatment or use of analgesic medication" as a measure for a relevant pain problem. Our data suggests that in addition to medication use, pain-related disability may be a key factor to judge pain severity. Such an approach has been implemented in the Chronic Pain Grading (*Von Korff et al., 1992*) and its pediatric adaption (*Wager et al., 2013b*) which includes pain intensity and pain-related disability to classify patients to different levels of pain severity.

Our results show that age and sex are significant albeit small predictors for health care utilization. Age was not a significant predictor in previous studies (*Perquin et al., 2001*; *Huguet & Miró, 2008*). These studies, however, included children with larger age ranges, 8 to 16 and 0 to 18, respectively (*Perquin et al., 2001*; *Huguet & Miró, 2008*). It may be that there is only a local association for age in the age range of this study sample (3–10 years). The reason why this was found is unknown. It may be due to the fact that pre-school children have more often a clearly identified medical condition or that parents

are more eager to find such an explanation for their children's pain. When looking at the association between age and health care utilization within a larger range, encompassing also adolescents, this effect may cease. However, it may also be, that the sample size in these studies was too low to detect this effect. Previous findings are conflicting with regard to sex; while sex did not have an influence on health care utilization in one study (*Huguet & Miró, 2008*), another study finds girls to be more likely to use health care (*Perquin et al., 2001*). Close inspection of the stability of our findings suggest that the effect of sex stabilizes extremely late, i.e., small sample sizes utilized in previous studies may be responsible for these inconsistencies.

Another significant predictor for health care utilization was a migration background. Children with a migration background were more likely to visit a physician. This difference may not be explained by ethnicity, but rather by effects of acculturation (*Chan, Hamamura & Janschewitz, 2013*). US-studies with adult chronic pain patients did not find an effect of ethnicity (*Meghani & Cho, 2009*). However, it was shown that a higher level of acculturation is associated with reduced pain threshold and pain tolerance (*Chan, Hamamura & Janschewitz, 2013*). We found no evidence for attending a physician for pain-treatment depending on SES. In contrast to this, access to *specialized* pain treatment seems to depend on parental SES (*Wager et al., 2013a*). In studies with adult chronic pain patients, results concerning the association between health care utilization due to chronic pain and SES are conflicting. While some report an increased utilization in patients with lower income (*Meghani & Cho, 2009*), others report increased utilization in those with a higher socioeconomic status (*Lim, Jacobs & Klarenbach, 2006*). Unlike adult pain patients, children with chronic pain and additional mental problems or low psychological well-being were not more likely to visit a physician (*Hirsch et al., 2014*). Neither psychological well-being nor behavioral problems are associated with visiting a doctor. Taken together we found that pain-specific measures were much more important to predict health care utilization in children with recurrent pain than demographic or generic measures.

On a methodological level, we were able to show that individual predictors in multiple logistic regression analysis may change their significance in samples even with several hundred participants. Sometimes effects that are significant in one direction change their direction later on. The sequential sampling approach taken here suggests that much larger sample sizes than employed in previous studies (*Perquin et al., 2001*; *Huguet & Miró, 2008*; *Toliver-Sokol et al., 2011*) are needed to substantiate claims about the significance of individual predictors. As demonstrated by the results concerning the effects of sex, this problem is especially severe when the significance or insignificance of small effects is assessed. This is due to the relationship between power, sample size, effect size and significance. Large effects require smaller sample sizes at a given power to be detected than small effects to become significant. Some effects that were significant at large sample sizes were simply too small to reach significance in smaller samples resulting in a larger $POS_{crit}$ for such small effects. As a result, $POS_{crit}$ can only be generalized to other samples if one assumes that the sample if drawn from a population with similar effect size. Similar relations

have been observed when the POS was determined for correlations. Here, small effects also required much larger sample sizes than large effects (*Schönbrodt & Perugini, 2013*).

Several recent other studies have used a similar approach to show that also tests for correlation coefficients and factor loadings need larger samples-sizes than traditionally believed before they yield reliable results (*Schönbrodt & Perugini, 2013*; *Hirschfeld, Von Brachel & Thielsch, 2014*). Solutions to these problems are developed within the accuracy in parameter estimation (AIPE) approach to reporting study results and sample size planning (*Kelley & Maxwell, 2003*). In contrast to traditional null-hypothesis significance testing approach (*Neyman & Pearson, 1928*) that exclusively focuses on whether or not a specific effect is null, or not, AIPE focuses on assessing estimates for effects with a given precision. This opens the possibility that while an effect may be significant in one study and non-significant in another, there is still the possibility that the parameter estimates in both studies overlap.

### Limitations

Several limitations need to be kept in mind when interpreting the results of the present study. Most of these are due to the fact that the data were from a large population-based survey that did not assess pain-related variables with the level of detail that is possible when a smaller group of patients are assessed at individual treatment centers (*Hirschfeld & Zernikow, 2013*). This is most relevant with regard to an assessment of the etiology of pain and the measure of pain-related disability. Based on the present data it is not possible to distinguish between primary and secondary pain problems, i.e., some children within this sample may report pain because of recurrent acute infections. At the same time we do not believe that this explains differences to earlier studies because only some excluded patients with known etiology (*Perquin et al., 2001*) while others do not differentiate (*Toliver-Sokol et al., 2011*). The presumably low stability of the results concerning pain-related disability cannot be explained by lack of reliability alone because the more comprehensive measures for HRQL and mental health problems did not yield more reliable effects. Nevertheless, future population-based studies should expand their focus on pain as a central variable, and utilize measures that balance the need for more detailed information on pain-related health care utilization and brevity.

### CONCLUSION

The aim of present study was to identify robust predictors of health care utilization in children with recurrent pain. We found that pain-related disability was by far the most important predictor for health care utilization, with children scoring high on this measure having a five-fold increased risk to attend a physician. Pain-intensity and frequency are also significant but of lesser relevance. This further supports the choice of this variable rather than pain-intensity as core-outcome in pediatric trials (*McGrath et al., 2008*). As a result, disagreements between studies may be due to chance alone. In order to substantiate specific claims about individual predictors we urge researchers to use sequential sampling techniques if the aim is to scrutinize the impact of sample size on the stability of results and comment on required sample sizes. If the aim is to the robustness of the results at

hand as an internal validation of the model (*Steyerberg et al., 2001*), we suggest using bootstrapping.

### Funding

The authors declare there was no funding for this work.

### Competing Interests

The authors declare there are no competing interests.

### Author Contributions

- G Hirschfeld conceived and designed the experiments, performed the experiments, analyzed the data, contributed reagents/materials/analysis tools, wrote the paper, prepared figures and/or tables, reviewed drafts of the paper.
- J Wager conceived and designed the experiments, performed the experiments, reviewed drafts of the paper.
- B Zernikow conceived and designed the experiments, analyzed the data, reviewed drafts of the paper.

### Human Ethics

The following information was supplied relating to ethical approvals (i.e., approving body and any reference numbers):

In this study the following data was used: Public Use File KiGGS, The German Health Survey for Children and Adolescents 2003–2006, Robert Koch Institute, Berlin (Germany), 2008.

Additional information about ethics approval is given in the method section.

### Data Deposition

The following information was supplied regarding the deposition of related data:

Data analysis was performed in R. Analysis scripts; the file "Public Use File KiGGS 2003-2006" and may be requested at: http://www.rki.de/EN/Content/Health_Monitoring/Public_Use_Files/application/application_node.html.

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
