# Peer review of "Physician consultation in young children with recurrent pain—a population-based study"

_PeerJ, doi:10.7717/peerj.916_

## Round 0.1 · original submission · Major Revisions

Please address all of the issues raised by reviewers 1 and 2 providing a point by point response as to how & where each issue was addressed in the re used manuscript. Please add information on the study characteristics and ensure the sequential sampling approach is clarified further.

·

Basic reporting

Very clearly reported with some minor exceptions as follows.

1. "pain-related impairment for each aspect on a scale ranging from “never” (=1) to “always” (=5)." Does "never" mean never impaired by pain, or never able to do the activity due to pain?

2. "Children were asked to judge their pain-related impairment" Previously it says parents made the judgments about pain. These questions are not only about impairment but about attribution of that impairment to pain -- these would be difficult questions especially for preschool-aged children. Please clarify for every measure whether it was self-report or parent-report, and for what ages if it was self-report.

3. Consider using "immigrant" instead of "migrated."

Experimental design

Excellent

Validity of the findings

1. "makes the difference between chronic pain as a clinically relevant condition and as a private phenomenon" ... "This finding underlines the need for a more holistic definition of clinically relevant chronic pain." These and other references to "chronic" pain are confusing as they imply that the pains evaluated in the survey were chronic. But the article states that the criteria were "the child had pain at least once in the past three month [sic]" and "the child visited a doctor due to his/her pain problem." Presumably many of the pains identified in this study were acute sore throats, muscle sprains, headaches, and the like. The placement of this study specifically in the context of recurrent or chronic pain seems hard to justify.

2. In the age range 3-10, younger subjects might be more likely than older children to have a common and clearly identified medical disorder accounting for their pain. This possible explanation of the age effect could be considered.

Additional comments

A valuable contribution to the literature, especially the emphasis on the sample size needed to come to valid conclusions about predictors of health-related behavior.

Reviewer 2 ·

Basic reporting

No Comments.

Experimental design

No Comments.

Validity of the findings

No comments.

Additional comments

This is an interesting study in a population-based study population of 2149 children aged 3-10 years from the German Kinder KiGGS study on predictors of whether a child attended a physician due to pain. Multiple logistic regression indicates that children with higher levels of pain-related disability, -intensity, -frequency as well as with a migration background and a female gender are significantly more likely to attend a physician. In an additional analysis the stability of the effect sizes in relation to the study size was analysed by a sequential sampling approach.

I have 3 general major comments:

1) The description of the procedure of the sequential sampling approach in the data analysis paragraph is not easy to follow. Please try to describe it in more detail. How exactly were the trajectories calculated? Concerning the definition of POS: How is the significance of an effect defined that should not change any more? What means “a trajectory stabilised”? How was POScrit calculated? Is it the 80th percentile?
2) Generally, please try to make more clear when results from a specific sampling order are described or those obtained from all 1000 repetitions of sampling.
3) The authors do not present characteristics of the study population apart from age and sex distribution. It is important to see the characteristics of the other analysed predictors such as SDQ, HRQL, migration, SES, pain variables,… as well. Preferably, the characteristics should be presented before transformation (scaling by division by 2sd).

Furthermore, there are some comments specific to certain parts of the manuscript:
Measures
4) Please add a very short description of KiGGS (design, age range, number of children originally recruited) instead of just referring other publications. Concerning the representativeness of the obtained results, does the study population in the current manuscript (less than half of the children from Kinder KiGGS) differ systematically from the original publication?
5) Please add more information on the range of the pain disability variable.
6) Please add 1-2 sentences about how the SDQ variable was calculated. The description in its current form is too brief and not readily understandable. Which version of the SDQ was used, the parent-reported or the self-reported version?
7) The SDQ is a screening questionnaire for behavioural problems. It assesses mental health problems and not “somatic symptoms”. Please use a different naming.
Data Analysis
8) Which statistical analysis programme was used?
Results
9) What is the correlation between the different pain variables?
10) Page 7: Please make more clear that the trajectories of a single one of the 1000 random sequential orders is described and discussed here for an explanation of the data analysis.
11) Page 7: It is no clear how it can be observed that “the effects of HRQL and SES were insignificant throughout the trajectory”. Please explain. To which trajectory is referred here? To the one shown in Figure 2?
Discussion
12) Page 10: “reliably determining the significance of individual predictors in multiple logistic regression affords extremely large data sets.”: “extremely” seems to be a too strong wording in this context.
13) Page 11: “This is in line with a recent studies that used this method to define the sample size necessary to stabilize correlation coefficients and factor loadings (Schönbrodt & Perugini, 2013; Hirschfeld, Brachel & Thielsch, 2014).“ It is not clear what this sentence means. Please explain.
Figures, Table:
1) Table 1: I assumed that the POS are specific to a certain sequential order of the participants. So each of the 1000 repetitions produced a set of POS. It is not clear to me to which of the 1000 repetitions the POS stated here correspond to. Do they refer to the example sampling order in figure 2?
2) The left hand side of Table 1 and Figure 1 contain the same information: once as numbers and once as a figure. Either the ORs in the table or the figure should be better moved to the supplement.
3) Table 1, Fig 1-3: Please give the figures and table more informative labels by adding a short description about the content, so that ideally the data in the table or the figure can be understood without further information from the text.
4) Figure 3: A horizontal line at 80% stabilized could be added in the figure to illustrate POScrit.

Finally, there are some minor comments:
1) Abbreviations should be defined in the text on first use and consistently used thereafter.
2) Page 3: Please correct: “once in the past three month” -> “once in the past three months”
3) Page 4: Please correct: “proceeding” into “preceding”.
4) Page 5: The second last sentence on this page is incomplete. Please check.
5) Labelling of the figures: Please change “Strength and difficulties questionnaire” to “Strengths and difficulties questionnaire”.
6) Page 8: “while the effect of HRQL seemed to stabilize with few participants in the specific order studied above, this effect stabilizes relatively late when alternative orders are considered”: To which effect of HRQL is referred here? Phy or Psy?
7) Page 9: Please correct: “with regards to” -> “with regard to”
8) Page 10: “In contrast to this access to specialized treatments seems to depend on parental SES”: Please add a comma after “this”.
9) Page 11: Please correct: “not significance” -> “insignificance”.

---

## Round 0.2 · Minor Revisions

Please address all of the issues raised by reviewer 2 and provide a point by point response as to how & where each issue was addressed in the revised manuscript.

·

Basic reporting

-

Experimental design

-

Validity of the findings

-

Additional comments

-

Reviewer 2 ·

Basic reporting

No comments.

Experimental design

No comments.

Validity of the findings

No comments.

Additional comments

Thank you for your considered responses to my comments and for preparing a revised manuscript. This revision substantially improved compared to the previous submission, especially with respect to the description of the sequential sampling approach.

However, several important points need to be mentioned:
- What exactly means that an effect “stabilized” (e.g. in the definition of POS= “number of participants that had to be included until the significance of this specific effect did not change any more or stabilized”).
- The stability of effects is strongly dependent on the effect size. Predictors with higher effect tend to stabilize earlier and those with lower effect size later. This can be explained by the power to detect an effect dependent on the sample size and effect size. Instead of repeatedly stating that effects stabilise at some point, it could be written that below a certain sample size the power to detect the effect was not big enough due to the effect size. In the current form, the reader might think, when he/she reads the effect stabilised (e.g. for sex) that this is then the N needed to come to a final conclusion about the effect. However, it might be that in another cohort with similar N no effect is observed due to the small effect size. Furthermore, a short explanation about the relationship between power, sample size, effect size and significance of effects might be added.
- The sentence in the discussion “An analysis of the variability of these results showed that large samples are needed to identify significant.” should be reformulated. What is large? Furthermore, the N to detect effects is dependent on the power/effect size.
- If the authors propose to the readers to apply sequential sampling approaches or bootstrapping, they might shortly describe the differences and explain why they chose the first method here.
- It is not clear what the authors want to express with the sentence “Furthermore, we demonstrated that multiple logistic regression analysis yields results that are much less stable than one may believe.” This sentence does imply that many researchers believe that this method yields stable results which I think is not correct. Thus, please remove this sentence.

Furthermore, several minor points remain:
Abstract:
- Please change “somatic symptoms” to “mental health problems”.
- Please make sure each abbreviation is written in full before first use.

Table 1:
- Please add a description to the first row to make clear whether “mean and sd” or “median and iqr” are stated for the continuous variables. If it is case 1, are all of the variables normally distributed? If not please provide “median and iqr”. So the description could be “mean +- sd / N(%)”.
- Please change “2-3 moth” to “2-3 per month”

Figure 1
- Please change “predicting whih children” to “predicting which children”.
- Please change “Strength and difficulties questionnaire” to “Strengths and difficulties questionnaire”

Figure 3:
- According to your response to my proposition to add a line at 80%, I assumed that this was planned to be done, but the figure was not updated.
- Please change “Strength and difficulties questionnaire” to “Strengths and difficulties questionnaire”.

---

## Round 0.3 · accepted · Accept

All issues have been adequately addressed to the satisfaction of the reviewers.

·

Basic reporting

-

Experimental design

-

Validity of the findings

-

Additional comments

-

Reviewer 2 ·

Basic reporting

-

Experimental design

-

Validity of the findings

-

Additional comments

Thank you for your considered responses to my comments. I have no more comments.